# Comparative efficacy of various oral hygiene care methods in preventing ventilator-associated pneumonia in critically ill patients: A systematic review and network meta-analysis

**Sachika Yamakita**[1], **Takeshi Unoki**[2]*, **Sachi Niiyama**[1,3], **Eri Natsuhori**[1,4], **Junpei Haruna**[5], **Tomoki Kuribara**[2]

1 Department of Acute and Critical Care Nursing, Graduate School of Nursing, Master's Program, Sapporo City University, Sapporo, Hokkaido, Japan, 2 Department of Acute and Critical Care Nursing, School of Nursing, Sapporo City University, Sapporo, Hokkaido, Japan, 3 Advanced Critical Care and Emergency Center, Sapporo Medical University Hospital, Sapporo, Hokkaido, Japan, 4 Intensive Care Unit, Sapporo Kojinkai Memorial Hospital, Sapporo, Hokkaido, Japan, 5 Intensive Care Unit, Sapporo Medical University Hospital, Sapporo, Hokkaido, Japan

* iwhyh1029@gmail.com

## Abstract

Oral hygiene care is important for ventilator-associated pneumonia prevention. However, the optimal oral hygiene care approach remains unclear. A network meta-analysis was conducted to compare the efficacy of various oral hygiene care methods for ventilator-associated pneumonia prevention in critically ill patients, and the methods were ranked. A literature search of three representative databases was conducted. We only analyzed parallel randomized controlled trials conducted to analyze the use antiseptics or toothbrushes in oral hygiene care for adult patients undergoing invasive mechanical ventilation in the intensive care unit. The outcome measure was the incidence of ventilator-associated pneumonia. Bias risk was assessed using the Cochrane Risk of Bias 2 tool, and the confidence in the evidence was evaluated using the CINeMA approach. Statistical analyses were performed using R 4.2.0., GeMTC package, and JAGS 4.3.1. The review protocol was registered in PROSPERO (registration number: CRD42022333270). Thirteen randomized controlled trials were included in the qualitative synthesis and twelve randomized controlled trials (2395 participants) were included in the network meta-analysis. Over 50% of the included studies were conducted in medical-surgical intensive care units. Ten treatments were analyzed and 12 pairwise comparisons were conducted in the 12 included studies. Analysis using surface under the cumulative ranking curves revealed that brushing combined with chlorhexidine 0.12% was most likely the optimal intervention for preventing ventilator-associated pneumonia (88.4%), followed by the use of chlorhexidine 0.12% alone (76.1%), and brushing alone (73.2%). Oral hygiene care methods that included brushing had high rankings. In conclusion, brushing combined with chlorhexidine 0.12% may be an effective intervention for preventing ventilator-associated pneumonia in critically ill patients. Furthermore, brushing may be the optimal oral hygiene care method for preventing ventilator-associated pneumonia in

**Data Availability Statement:** All relevant data are within the manuscript and its Supporting Information files.

**Funding:** This work was funded by ALCARE CO., Ltd (grant number: N/A) and TU was received. https://www.alcare.co.jp/en/ The funders had no role in the study design, data collection and analysis, decision to publish, or preparation of the manuscript.

**Competing interests:** The authors have declared that no competing interests exist.

the intensive care unit. Further research is needed to verify these findings as the CINeMA confidence rate was low for each comparison.

## Introduction

Oral hygiene care is an important strategy for preventing ventilator-associated pneumonia (VAP). VAP is one of the most common hospital infections in mechanically ventilated patients in the intensive care unit (ICU) and is associated with unfavorable outcomes [1–5]. Tools, such as toothbrushes and swabs, and solutions such as chlorhexidine (CHX), water, and povidone-iodine, are used for providing oral hygiene care in clinical practice [6–9]. These may be used alone or in combination.

Toothbrushing is recognized as an important component of oral hygiene care in critically ill patients allowing physical removal of dental plaque that is a potential nidus of infection [10]. A network meta-analysis (NMA) comparing 16 oral hygiene interventions has demonstrated that toothbrushing alone is the most effective approach [11]. However, important limitations of the NMA are the inclusion of studies with a duration of intervention that was shorter than the use of mechanical ventilation and of studies including interventions directed to the pharynx rather than the oral cavity [8,12–14]. Furthermore, toothbrushes are a low-cost intervention with high feasibility for implementation. In surveys investigating oral care methods used in ICUs, toothbrushes are frequently utilized [15,16]. Thus, toothbrushes are considered a relatively easy-to-apply and effective method for preventing VAP in clinical settings.

Therefore, we hypothesized that oral hygiene care involving toothbrushing would be the most effective method for preventing VAP and conducted an NMA to compare the efficacy of various oral hygiene care methods, including the use of antiseptics and toothbrushing, on the incidence of VAP in adult patients undergoing invasive mechanical ventilation. In addition, we ranked these methods by effectiveness for practical consideration.

## Methods

### Protocol and registration

The protocol for this study was registered in the International Prospective Register of Systematic Reviews (PROSPERO) in May 2022 (CRD42022333270). This study was conducted and reported in accordance with the Preferred Reporting Items for Systematic Reviews and Meta-Analyses (PRISMA) extension statement for Network Meta-Analyses [17].

### Eligibility criteria

**Population.** Participants were adult patients who were admitted to the ICU and underwent invasive mechanical ventilation.

**Intervention.** Oral hygiene care using antiseptics and/or tooth brushing was provided continuously throughout the duration of mechanical ventilation. Studies were excluded if any of the following criteria applied: (1) oral hygiene care included the use of antibiotics; (2) patients underwent an intervention that directly involved the larynx and/or pharynx; (3) oral hygiene care was provided temporarily and/or not continuously (e.g., only once after tracheal intubation, for three days only regardless of whether ventilation was required for longer); (4) the intervention involved the use of complex care bundles in which the differences between

the intervention and control groups included factors other than tooth brushing, rinsing the mouth, or applying gel (e.g., differences in body position, modification of tube cuff pressure).

Regarding eligible treatments included in the treatment network, CHX was stratified according to its concentration. Manual and electronic brushing were considered the same treatment. Different frequencies of the same intervention were considered the same node. Different types of antiseptics were considered the same treatment because stratification of antiseptics according to type led to several categories of treatments, which may affect the network geometry.

**Comparison and control.**   Comparisons were usual care (e.g., using only normal saline), placebo, and oral hygiene intervention care if the studies met the inclusion criteria. In this NMA, the control was described as placebo or usual care, with no tooth brushing or use of antiseptics (e.g., using only normal saline).

**Outcome and study design.**   The outcome was the incidence of VAP. The definition of VAP was based on the definition used in each study. Only parallel randomized controlled trials were included. Articles without full text, letters without reported data, and reviews were excluded.

## Search strategy and study selection

A previously described comprehensive search strategy was employed for the identification of eligible articles published in the following databases: Cochrane Central Register of Controlled Trials (CENTRAL) via Ovid, MEDLINE via PubMed, and CINAHL via EBSCO. The search strategy is presented in the S1 Text. The International Clinical Trials Registry Platform was also searched for the identification of registered ongoing trials. The databases were searched from inception to March 24, 2024. Manual searches were also conducted to identify additional relevant studies. We manually checked all the references lists of the relevant systematic reviews to identify any additional studies. Selection of articles was not limited to a specific language.

Two of the five reviewers (SY, EN, SN, JH, and TU) independently screened the titles and abstracts of the articles extracted using the search strategy and identified potentially relevant studies. Thereafter, two of the reviewers independently assessed the eligibility of the screened articles by reviewing their full texts. Disagreements were resolved through discussion and arbitration by another author (TU) when necessary. Finally, the eligible studies were included in the qualitative synthesis and NMA.

## Data collection

Two authors (SY and SN) independently extracted data from each included study using a specially designed data extraction form. Disagreements were resolved through reconfirmation and discussion.

The extracted data included the characteristics of the study (i.e., title, author, year of publication, location of the trial, and type of ICU), participant details (i.e., age, sample size, sex, and severity of illness score), data on oral hygiene care (i.e., type of antiseptics, concentration of solutions, use of toothbrushing), and reported outcomes (i.e., incidence of VAP).

## Risks of bias within studies

The risk of bias in each study was independently assessed at the outcome level by two authors (SY and TU) using the Cochrane Risk of Bias tool 2.0 [18], which comprises the following five domains: (1) bias arising from the randomization process; (2) bias due to deviations from intended interventions; (3) bias due to missing outcome data; (4) bias in the measurement of the outcome; (5) and bias in the selection of the reported result. Risks of bias were assessed as

"low," "high," or "some concerns," and the overall risk of bias was judged based on the results for the domains. Conflicts were resolved through discussions and consensus.

## Statistical analysis

We conducted the NMA using the Bayesian approach. NMA can obtain direct and indirect evidence of a correlation between treatment effects observed in various comparisons conducted in multiple trials [19,20]. For the analysis of multi-arm trials with more than two intervention arms, each study was included as a two-arm comparison series in the dataset. Thus, we obtained the treatment effects for each intervention in relation to every comparator in the multi-arm. We adjusted the data using the "powerAdjust" argument, an option available in the GeMTC package, setting the weight for multi-arm trials at 0.7 for correlations between arms within the same study [21–23]. The analysis was performed using the Markov-Chain Monte-Carlo (MCMC) method. The NMA was performed using a random effects model. To ensure convergence of the Markov chains, we discarded the first 5,000 iterations as the burn-in period and ran four chains in parallel. The convergence was assessed using two diagnostics, the visual checking of trace plots and the potential scale reduction factor (PSRF) of Gelman-Rubin. Statistical analyses were performed using R 4.2.0 (R Foundation for Statistical Computing, Vienna, Austria)., GeMTC package, and JAGS 4.3.1 (Source Forge, California, United States of America).

Dichotomous variables are presented as risk ratios with 95% confidence intervals (CI) or credential intervals (CrI). Continuous variables are presented as mean and standard deviation or median and interquartile range. The results of comparison of the treatments were summarized in a forest plot by setting oral care without either an antiseptic or toothbrushing as a reference treatment and calculating the risk ratio. The ranking of the treatments was analyzed using the surface under the cumulative ranking curve (SUCRA). The SUCRA is represented as a percentage and indicates the relative probability that one treatment is the best option [24,25]. For discrete variables, we conducted imputation of missing values as worst–worst scenario.

In the network plot, each intervention analyzed in the studies was classified as a treatment and indicated using nodes. Comparisons between treatments are indicated using links connecting the nodes. The size of the node was weighted based on the number of studies that involved the analysis of the relevant interventions. The thickness of the link indicates the number of studies that involved the analysis of the two interventions at either end of the link.

Overall consistency in the network was evaluated using I2 statistics, with values >40% were considered indicative of substantial heterogeneity [26]. The consistency of treatment effect estimates between direct and indirect comparisons was evaluated using the node-splitting method. The results of the node-splitting analysis were presented in a forest plot and used for calculating p-values; therefore, we confirmed the overlapping of 95% CIs through visual assessment [19]. P < 0.05 was considered statistically significant.

## Assessment of the quality of the evidence using Confidence in Network meta-analysis (CINeMA)

Confidence in the evidence was evaluated using the CINeMA approach [27,28]. CINeMA is broadly based on the GRADE framework and covers 6 domains: (i) within-study bias (referring to the impact of risk of bias in the included studies), (ii) reporting bias (referring to publication and other reporting biases), (iii) indirectness, (iv) imprecision, (v) heterogeneity, and (vi) incoherence. Each of these domains can assessed according to three levels of concern (no concerns, some concerns, or major concerns) and the reviewer's input is required at the study level for within-study bias and indirectness. Judgments across domains are summarized to

obtain 4 levels of confidence for each relative treatment effect, corresponding to the usual GRADE assessments of very low, low, moderate, or high. Two authors (SY and TU) independently made judgments about the quality of the evidence. Conflicts were resolved through discussions and consensus. These assessments were conducted using the CINeMA web application [29].

### Additional analyses

Publication bias was determined by visually assessing a funnel plot. In addition, we conducted a sensitivity analysis of VAP by excluding studies in which assessors were not blinded because blinding of the VAP assessor was associated with the results.

## Results

### Study selection

We initially identified 6334 articles in the databases, 34 records in the registry, and three articles in the manual search of previous systematic reviews (Fig 1). After excluding duplicate articles, we screened the titles/abstracts of the remaining 4477 articles and 34 records. After the screening, 4269 irrelevant articles were excluded, and four articles were not retrieved. The full texts of the remaining 238 articles were reviewed, and 225 studies were excluded for various reasons, such as differences in study design, inclusion of non-intubated patients, duration of

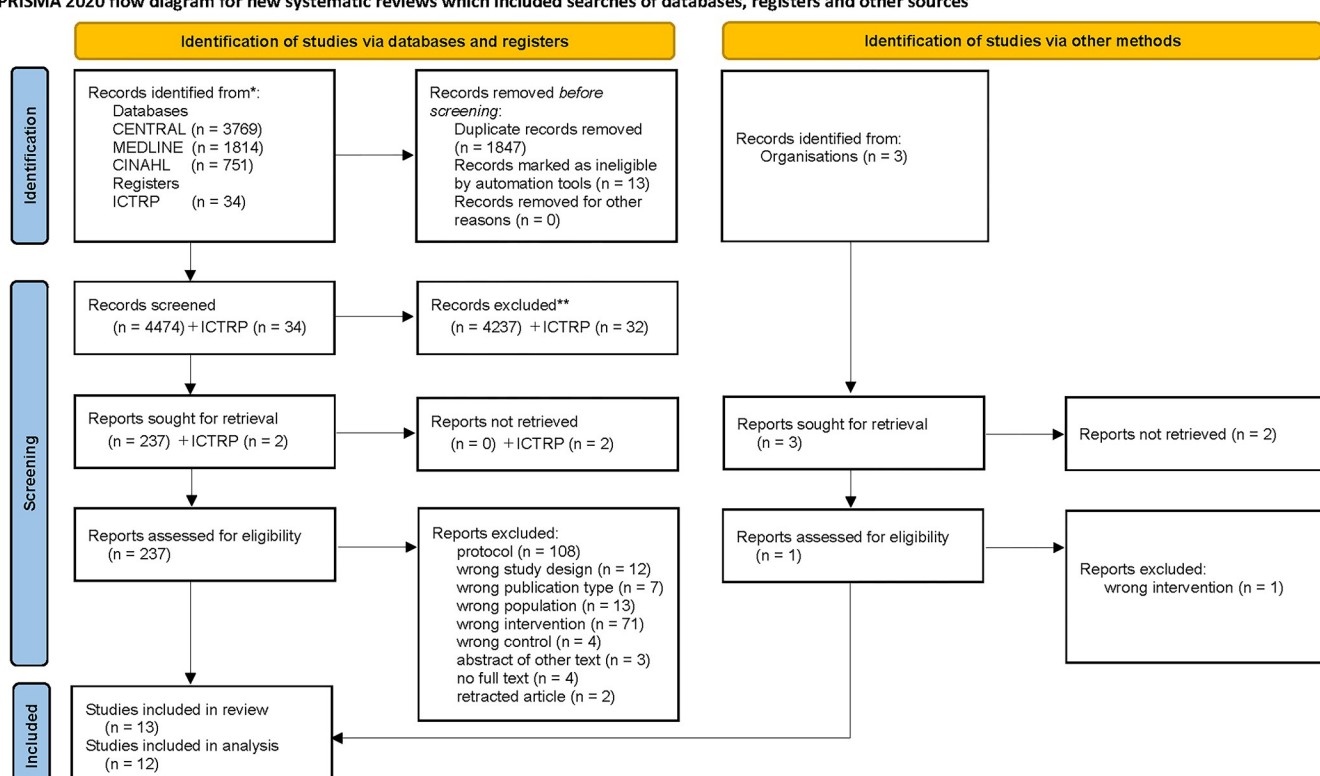

PRISMA 2020 flow diagram for new systematic reviews which included searches of databases, registers and other sources

*Consider, if feasible to do so, reporting the number of records identified from each database or register searched (rather than the total number across all databases/registers).
**If automation tools were used, indicate how many records were excluded by a human and how many were excluded by automation tools.

*From:* Page MJ, McKenzie JE, Bossuyt PM, Boutron I, Hoffmann TC, Mulrow CD, et al. The PRISMA 2020 statement: an updated guideline for reporting systematic reviews. BMJ 2021;372:n71. doi: 10.1136/bmj.n71. For more information, visit: http://www.prisma-statement.org/

**Fig 1. PRISMA flow diagram.**

intervention shorter than the duration of mechanical ventilation, use of complex care bundles, and study report was the abstract of other literature. Finally, 13 studies were included in the qualitative synthesis and 12 studies were included in the network meta-analysis.

## Characteristics of the included studies

The characteristics of the studies included in the qualitative synthesis are summarized in Table 1. The included studies were published between 2000 and 2024. More than half of the studies were conducted in medical-surgical intensive care units [30–35] and two were multi-center studies [36,37]. Of the 13 studies included, 10 were two-arm studies [30,31,33,34,36–41] and three were three-arm studies [32,35,42]. We integrated the interventions in one of the three-arm studies because the only difference between the interventions was frequency [42]. The combined sample size of the 12 included studies in the network meta-analysis was 2395 participants.

## Summary of network geometry

Fig 2 shows the network plot of the comparisons of the outcomes of the different treatments. Ten treatments were analyzed and 12 pairwise comparisons were conducted in the 12 included studies. There were no missing values in the extracted data. In the network plot, each treatment is indicated as blue dots (nodes), and the size of the dots indicates the total number of participants who underwent the intervention. Comparisons between treatments are indicated as black lines (links), and the thickness of the lines indicates the number of studies in which two treatments were analyzed. Controls are described as placebo or usual care, which was defined as not using brushing and/or antiseptics.

The largest node was "brushing only" and "brushing combined with CHX 0.12%," which was analyzed in four studies, followed by "brushing combined with CHX 0.2%," "CHX 0.12% only," and "control group," which were analyzed in three studies. Of the twelve links, the most common direct comparison was "brushing combined with CHX 0.12%" and "CHX 0.12% only," which was compared in three studies.

In the networks for comparisons of multiple treatment, the links between nodes indicate evidence of direct comparisons between interventions. The thickness of the links indicates the number of studies in which the two treatments were analyzed, and the size of the nodes indicates the total number of participants who underwent the intervention. Controls (Ctr) received placebo or usual care, which was defined as not using brushing and/or antiseptics.

Br, brushing only; BrBicarbonate, brushing combined with bicarbonate; BrCHX012, brushing combined with chlorhexidine 0.12%; BrCHX02, brushing combined with chlorhexidine 0.2%; BrCHX2, brushing combined with chlorhexidine 2%; BrListerine, brushing combined with Listerine; CHX012, chlorhexidine 0.12% only; CHX02, chlorhexidine 0.2% only; CHX2, chlorhexidine 2% only; Ctr, control group.

The sources of data for each intervention: Br [32,33,35,42], BrBicarbonate [32,35], BrCHX012 [31,34,40,42], BrCHX02 [32,38,39], BrCHX2 [33,39], BrListerine [35], CHX012 [31,34,40], CHX02 [30,36,38], CHX2 [37], Ctr [30,36,37].

## Risks of bias within studies

The results of the risk of bias assessment of the included studies are shown in Fig 3 [green (+): low risk, red (x): high risk, and yellow (-): some concerns]. Of the 12 included studies, six were deemed to have a low risk of bias [31,34,36,39,40,42], five had some bias concerns [30,32,33,35,37], and one had a high risk of bias [38]. Some of the studies had 'some concerns' in the overall risk of bias because we could not assess the "bias in the selection of the reported

**Table 1. Characteristics of the studies included in the qualitative synthesis.**

| Reference, year | Country | Method of oral hygiene care | Number of participants | Age [b] (year) | Males (%) | Severity of illness score [b] | Type of ICU | Blinding of VAP assessor |
|---|---|---|---|---|---|---|---|---|
| Fourrier, 2000 [30] | France | Chlorhexidine (0.2%) | 30 | 51.2±15.2 | 19 (63.3) | 37±15 [g] | Medical-surgical ICU | Yes |
| | | Placebo/usual | 30 | 50.4±15.5 | 19 (63.3) | 33±13 [g] | | |
| Fourrier, 2005 [36] | France | Chlorhexidine (0.2%) | 114 | 61.0±14.7 | 83 (72.8) | 45.2±17.5 [g] | Not reported (six ICUs, six hospitals) | Yes |
| | | Placebo/usual | 114 | 61.1±14.9 | 73 (64.0) | 45.0±17.5 [g] | | |
| Koeman, 2006 [37] | Netherlands | Chlorhexidine (2%) | 127 | 60.9±15.3 | 66 (52) | 22.2±7.02 [e] | Five mixed ICUs and two surgical ICUs (seven ICUs, five hospitals) | Yes |
| | | Placebo/usual | 130 | 62.1±15.9 | 93 (72) | 21.8±7.43 [e] | | |
| Scannapieco, 2009 [a] [42] | USA | Toothbrushing +chlorhexidine (0.12%) | I1 = 58 | 44.8±19.9 | 43 (74) | 18.5±4.1 [f] | Trauma ICU | Yes |
| | | | I2 = 58 | 47.6±19.1 | 44 (75.9) | 19.7±6.1 [f] | | |
| | | Toothbrushing | 59 | 50.0±22.5 | 36 (61) | 19.1±6.1 [f] | | |
| Pobo, 2009 [31] | Spain | Toothbrushing +chlorhexidine (0.12%) | 74 | 55.3±17.9 | 49 (66.2) | 18.8±7.1 [e] | Medical-surgical ICU | Yes |
| | | Chlorhexidine (0.12%) | 73 | 52.6±17.2 | 46 (63.0) | 18.7±7.3 [e] | | |
| Berry, 2011 [32] | Australia | Toothbrushing +bicarbonate | 33 | 60.4±17.5 | 42 (55.3) | 22.0±7.5 [f] | Medical-surgical ICU | Yes |
| | | Toothbrushing +chlorhexidine (0.2%) | 33 | 58.2±19.4 | 35 (49.3) | 22.8±7.8[f] | | |
| | | Toothbrushing | 43 | 59.1±18.1 | 44 (56.4) | 21.64±7.8[f] | | |
| Meinberg, 2012 [33] | Brazil | Toothbrushing +chlorhexidine (2%) | 28 | 40.1±14.6 | Not reported | 17.9±4.5 [e] | Medical-surgical ICU | Yes |
| | | Toothbrushing | 24 | 41.0±19.0 | | 16.7±6.8 [e] | | |
| Lorente, 2012 [34] | Spain | Toothbrushing +chlorhexidine (0.12%) | 217 | 61.0±15.6 | 146 (67.3) | 17.88±8.84 [e] | Medical-surgical ICU | Yes |
| | | Chlorhexidine (0.12%) | 219 | 60.4±16.6 | 145 (66.2) | 19.16±9.88 [e] | | |
| Berry, 2013 [35] | Australia | Toothbrushing +bicarbonate | 133 | 54.93±19.5 | 79 (59.4) | 21.38±8.0 [f] | Medical-surgical ICU | Yes |
| | | Toothbrushing +Listerine | 127 | 59.96±18.0 | 73 (57.5) | 21.21±8.0 [f] | | |
| | | Toothbrushing | 138 | 58.82±16.7 | 84 (60.9) | 20.86±7.7 [f] | | |
| Chacko, 2017 [38] | India | Toothbrushing +chlorhexidine (0.2%) | 104 | 41.02±17.78 | 45 (44.1) | Not reported | Medical ICU | No |
| | | Chlorhexidine (0.2%) | 102 | 45.91±18.38 | 70 (67.3) | | | |
| Zand, 2017 [39] | Iran | Toothbrushing +chlorhexidine (2%) | 57 | 45.43±2.95 | (80.7) [c] | 61.33±2.54 [d] | Trauma, surgery, neurosurgery, and general ICUs | Yes |
| | | Toothbrushing +Chlorhexidine (0.2%) | 57 | 44.45±2.72 | | 56.01±2.46 [d] | | |
| Lacerda, 2017 [40] | Brazil | Toothbrushing +chlorhexidine (0.12%) | 105 | 59.4±14.5 | 51 (48.6) | 21.9±7.5 [e] | Clinical/surgical and cardiology ICUs | Yes |
| | | Chlorhexidine (0.12%) | 108 | 63.2±14.5 | 54 (50.0) | 22.2±7.7 [e] | | |
| Santos, 2024 [41] | Brazil | Chlorhexidine (0.12%) | 45 | 63 | 25 (55.0) | Not reported | general medicine ICUs | None |
| | | Toothbrushing +chlorhexidine (0.12%) | 45 | 65 | 31 (69.0) | | | |

ICU, Intensive care unit; VAP, ventilator-associated pneumonia; APACHE, Acute Physiology and Chronic Health Evaluation; SAPS, Simplified Acute Physiological Score. Placebo/usual was defined as not using brushing and/or antiseptic.

[a]Although this was a three-arm study, some interventions were integrated because the only difference between them was frequency.

[b]Age and severity of illness scores are reported as mean ± standard deviation.

[c]Percentage of males in the study population; the authors reported no significant difference in each group.

[d]APACHEIV.

[e]APACHEII.

[f]APACHE.

[g]SAPSII.

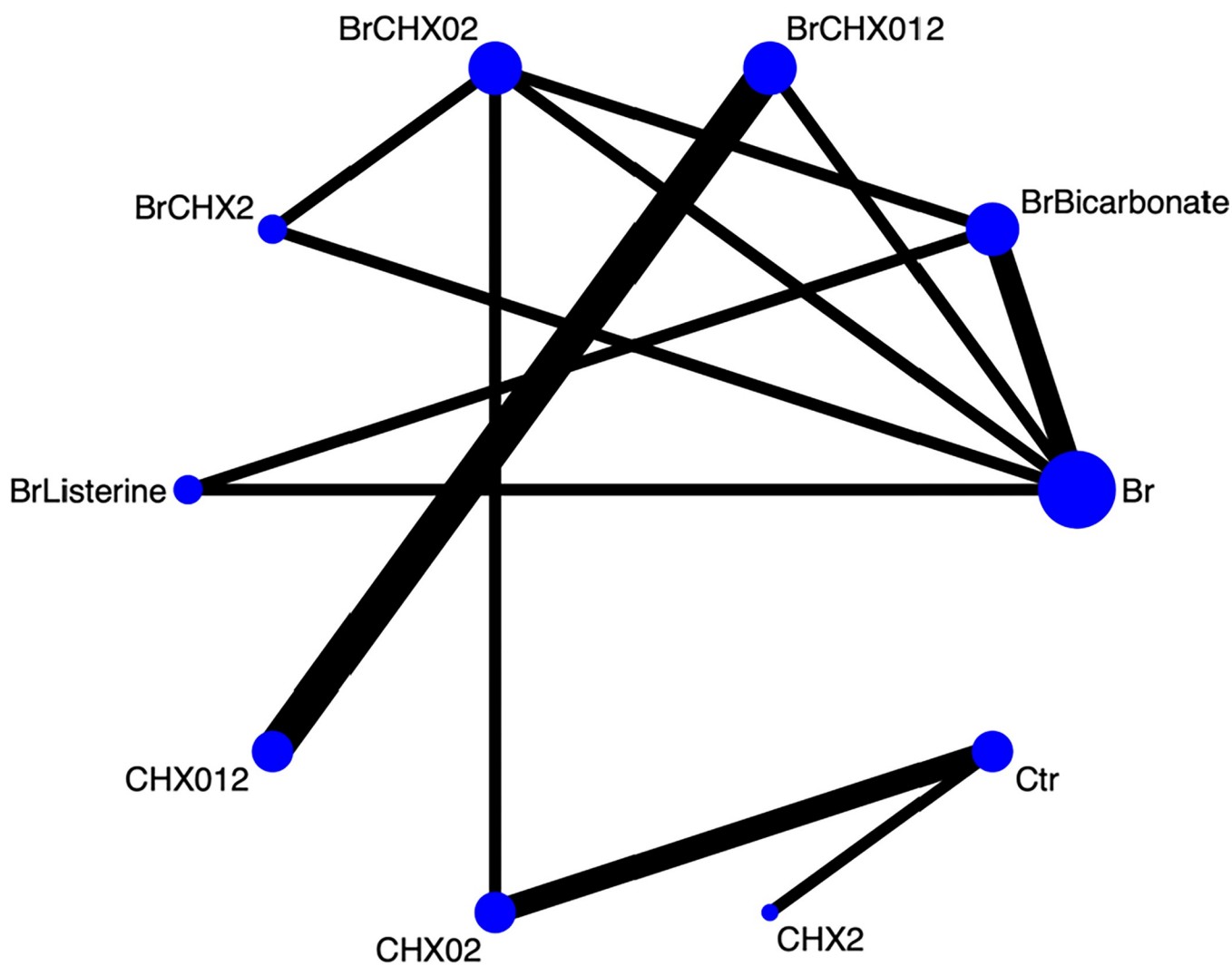

**Fig 2. Network plot of oral hygiene care.**

result" domain owing to the lack of a study protocol. Moreover, we noted that one study [32] had possible deviations because of its trial context; however, there was no evidence that these deviations affected the reported outcome. Another study was deemed to have a high risk of bias in the "bias in the measurement of the outcome" domain because nurses may have been involved in the diagnosis of VAP; however, the nurses were not blinded to the intervention groups [38].

### Exploration for inconsistency

Direct and indirect comparisons of the consistency of the treatment effect estimates using the node-splitting method did not reveal any significant inconsistency (all $p > 0.05$, S1 Fig). The heterogeneity of the overall network was calculated using the $I^2$ statistic and no significant heterogeneity was observed ($I^2 = 7\%$). The network meta-analyses were performed using a random effects model. The PSRF value was 1 and it stabilized after 100,000 iterations, confirming that the convergence effect was good.

**Fig 3. Risk of bias assessment of the included studies.**

### Synthesis of results and confidence in NMA

Fig 4 shows the risk ratio for each intervention compared with the control. Compared to the control intervention, brushing with CHX 0.12% (RR = 0.05; 95% Crl, 0.00–0.93; certainty of evidence: low) significantly reduced the incidence of VAP.

The results obtained from the SUCRA analysis are presented in Fig 5 and Table 2. The SUCRA revealed that brushing combined with CHX 0.12% was most likely to be ranked as the optimal intervention (88.4%), followed by CHX 0.12% only and brushing only (76.1% and 73.2%, respectively). The lowest ranking group was the control group, which used neither antiseptics nor toothbrushing. Methods that involved brushing occupied the first and the third to

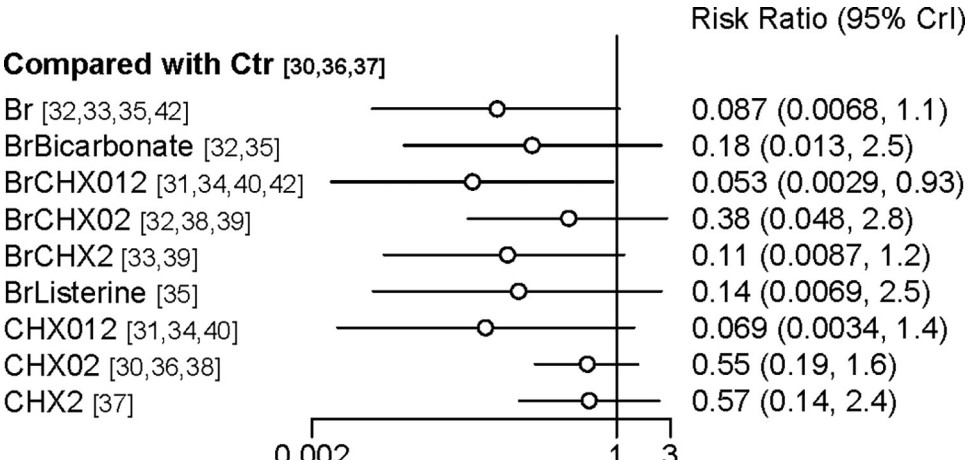

**Fig 4. Forest plot of efficacy the treatments in preventing VAP compared with the control.** Br, brushing only; BrBicarbonate, brushing combined with bicarbonate; BrCHX012, brushing combined with chlorhexidine 0.12%; BrCHX02, brushing combined with chlorhexidine 0.2%; BrCHX2, brushing combined with chlorhexidine 2%; BrListerine, brushing combined with Listerine; CHX012, chlorhexidine 0.12% only; CHX02, chlorhexidine 0.2% only; CHX2, chlorhexidine 2% only; Ctr, control group.

seventh ranks, whereas CHX 0.12% only was ranked second. However, the confidence rating with CINeMA was low or very low in each comparison. The main reasons for downgrading were related to within-study bias, with many studies rated as having 'Some concerns' due to the absence of protocols. Additionally, concerns regarding imprecision were noted due to the small number of included studies. The results of the CINeMA assessment are provided in the Supporting Information (S2 Fig and S3 Table).

## Results of additional analyses

The funnel plot is shown in the S3 Fig. The likelihood of publication bias was considered to be low because no obvious asymmetry was observed on visual assessment.

The VAP assessor in one of the 12 included studies was not blinded [38]; therefore, we attempted a sensitivity analysis. However, the direct comparisons were insufficient for connecting the networks if the applicable study was excluded; thus, the sensitivity analysis could not be conducted. Fig 6 shows the network plot for a scenario in which the network was not connected.

## Discussion

In this study, we hypothesized that oral hygiene care with toothbrushing would be the most effective method for preventing VAP in ICU patients, so we conducted an NMA to compare and rank the efficacies of various oral hygiene care methods for the prevention of VAP. The results showed that brushing combined with CHX 0.12% was most likely to be selected as the optimal oral hygiene care intervention for preventing VAP in ICU patients, followed by CHX 0.12% only and brushing only. In addition, the results indicated that the probability rankings for CHX 0.12% only and brushing only were similar; however, oral hygiene care methods that included brushing ranked higher overall.

The ranking of oral hygiene care methods for preventing VAP in the present study differs from that reported in previous studies. Sankaran et al. compared 16 oral hygiene care methods analyzed in 25 studies and found that the three best interventions were brushing only,

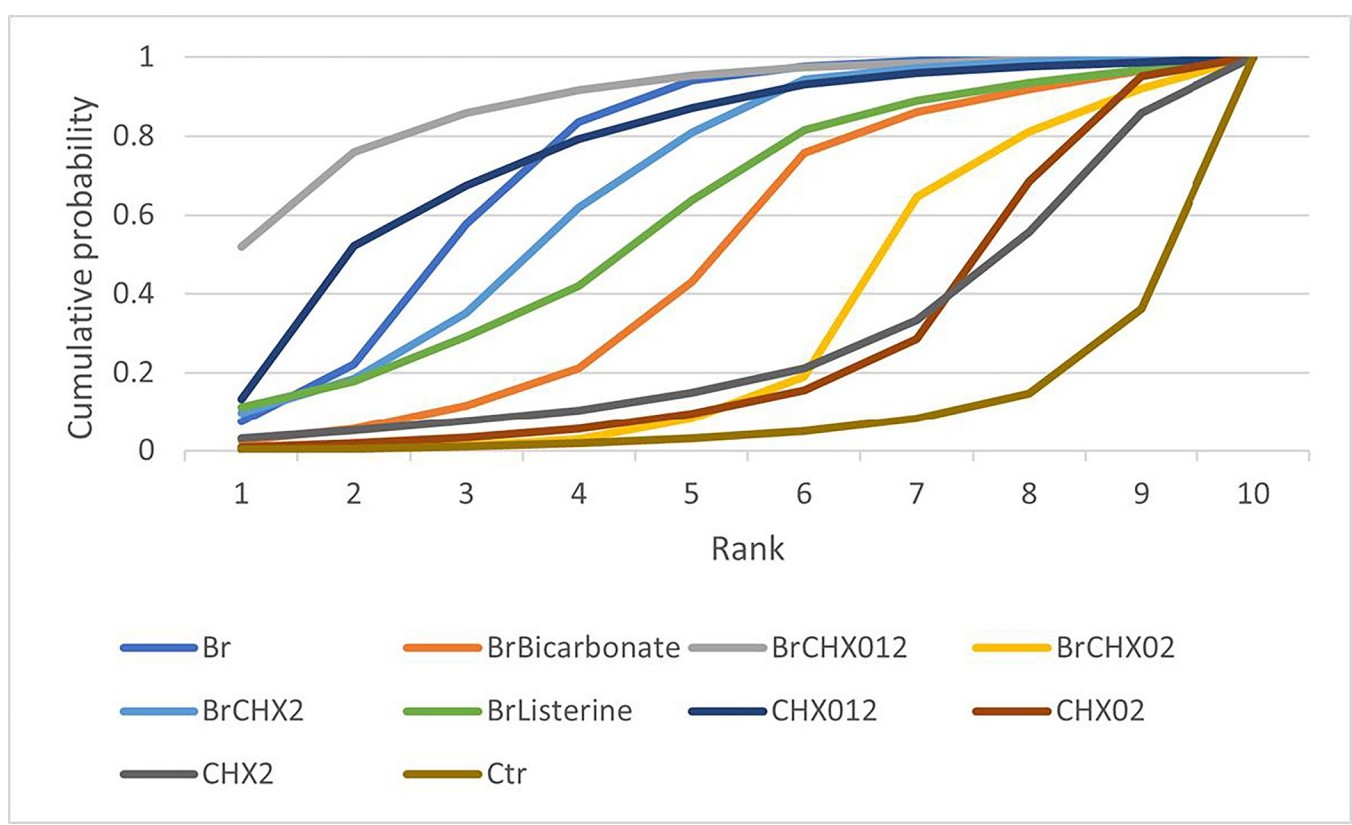

**Fig 5. The surface under the cumulative ranking curve (SUCRA) of the efficacy of oral hygiene care interventions in reducing the incidence of VAP.** The SUCRA value is the probability that each intervention is the best in the network. Larger values represent probabilities of higher ranking. Br, brushing only; BrBicarbonate, brushing combined with bicarbonate; BrCHX012, brushing combined with chlorhexidine 0.12%; BrCHX02, brushing combined with chlorhexidine 0.2%; BrCHX2, brushing combined with chlorhexidine 2%; BrListerine, brushing combined with Listerine; CHX012, chlorhexidine 0.12% only; CHX02, chlorhexidine 0.2% only; CHX2, chlorhexidine 2% only; Ctr, control group. The sources of data for each intervention: Br [32,33,35,42], BrBicarbonate [32,35], BrCHX012 [31,34,40,42], BrCHX02 [32,38,39], BrCHX2 [33,39], BrListerine [35], CHX012 [31,34,40], CHX02 [30,36,38], CHX2 [37], Ctr [30,36,37].

brushing with povidone-iodine, and use of furacillin [11]. Yang et al. classified the efficacies of different concentrations of CHX (0.12%, 0.2%, and 2%) for oral hygiene reported in 19 studies, and compared four methods used in the placebo groups. The authors reported that CHX 2%

**Table 2. Ranking order of the treatments based on the SUCRA analysis.**

| Method of oral hygiene care | SUCRA |
|---|---|
| Brushing with chlorhexidine (0.12%) | 0.88353889 |
| Chlorhexidine (0.12%) | 0.76095000 |
| Brushing | 0.73189444 |
| Brushing with chlorhexidine (2%) | 0.65582778 |
| Brushing with Listerine | 0.58651667 |
| Brushing with bicarbonate | 0.49239444 |
| Brushing with chlorhexidine (0.2%) | 0.30135556 |
| Chlorhexidine (2%) | 0.25947222 |
| Chlorhexidine (0.2%) | 0.25032222 |
| No brushing or antiseptics (Control) | 0.07772778 |

Legends: SUCRA, Surface Under the Cumulative Ranking Curves.

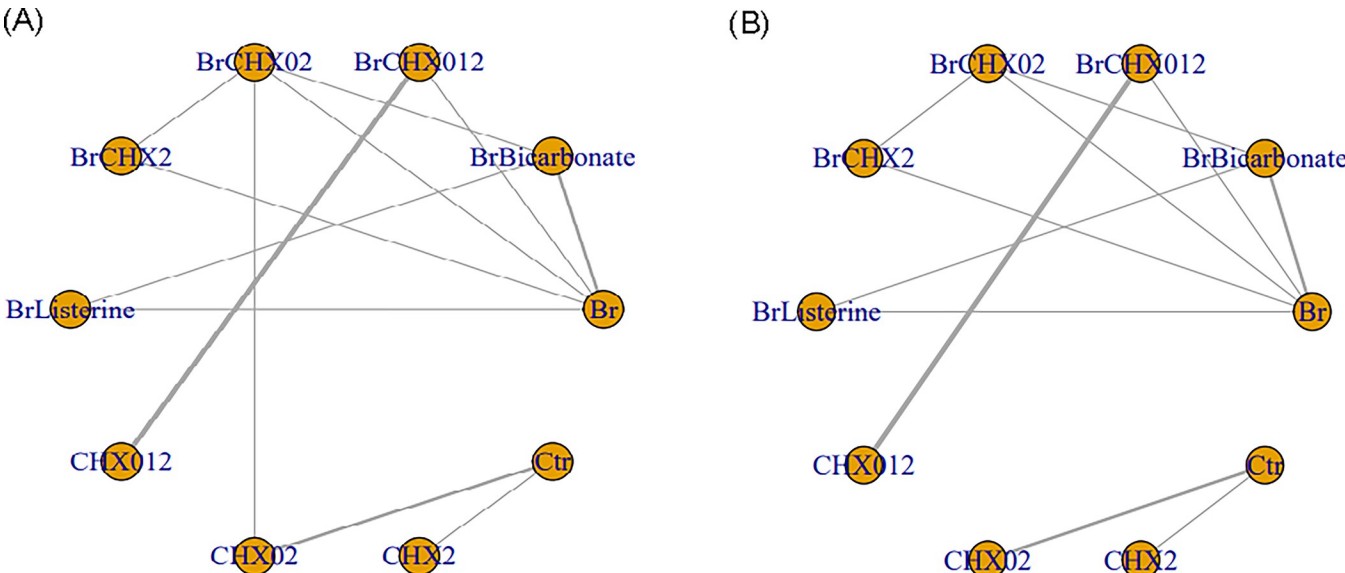

**Fig 6. Comparison of the network plot before and after excluding the applicable literature to conduct the sensitivity analysis.** Compared with the network plot created in this study (A), the network plot used for the sensitivity analysis had a gap that was not connected to the network (B). Br, B = brushing only; BrBicarbonate, brushing combined with bicarbonate; BrCHX012, brushing combined with chlorhexidine 0.12%; BrCHX02, brushing combined with chlorhexidine 0.2%; BrCHX2, brushing combined with chlorhexidine 2%; BrListerine, brushing combined with Listerine; CHX012, chlorhexidine 0.12% only; CHX02, chlorhexidine 0.2% only; CHX2, chlorhexidine 2% only; Ctr, control group. The sources of data for each intervention: Br [32,33,35,42], BrBicarbonate [32,35], BrCHX012 [31,34,40,42], BrCHX02 [32,38,39], BrCHX2 [33,39], BrListerine [35], CHX012 [31,34,40], CHX02 [30,36,38], CHX2 [37], Ctr [30,36,37].

and CHX 0.12% may be optimal oral hygiene care methods for preventing VAP, with CHX 2% ranking higher than CHX 0.12% [43]. The differences in rankings between the present study and previous studies may be attributed to two main reasons. First, there were differences in the inclusion criteria between our study and previous studies. Although Sankaran et al. conducted an NMA, the authors included studies in which the duration of intervention was shorter than that of mechanical ventilation and studies that with interventions directed to the pharynx rather than the oral cavity [8,12–14]. Although studies with these characteristics were excluded from the present study because they did not meet the inclusion criteria, some of the studies also included the evaluation of interventions such as use of povidone-iodine only, brushing with povidone-iodine, and furacillin; therefore, these methods of oral hygiene care were not included in the present study. Yang et al. conducted an NMA of studies on the use of different CHX concentrations for oral hygiene care; however, the authors did not consider whether brushing was included in the intervention [43]. In the present study, we classified not only differences in CHX concentration but also whether use of CHX was combined with brushing; therefore, the ranking of the methods differed from the previously reported ranking. Second, the present study included the most recent research in the field. Sankaran et al. analyzed the literature included in a Cochrane review published in 2016 [11,44], which was subsequently updated in 2020 [9]. In the present study, we included three articles published in 2017, including reports of studies in which the efficacy of different CHX concentrations and use of CHX with or without brushing were compared [38–40]. Therefore, the more recent studies included in the present NMA may have directly and indirectly added evidence that affected the rankings.

The use of a toothbrush for oral hygiene care may be effective in preventing VAP. In the present study, oral hygiene care methods that include brushing tended to rank high among the ten oral hygiene care methods analyzed. The results of meta-analysis conducted by Cochrane

Review suggested the efficacy of brushing for preventing VAP [9]. Additionally, an RCT that included 3,260 participants indicated that the efficacy of brushing alone was not significantly different from that of brushing with 0.12% CHX in preventing infection-related ventilator-associated complications, including VAP [45]. In other words, the significance of using CHX 0.12% for oral hygiene care was analyzed in the study, and the efficacy of brushing was demonstrated. However, a step-wedge cluster design was applied in this RCT; thus, we could not include it in the present study. It should be noted that this may have influenced the rankings in the present study by elevating the efficacy of brushing in our results. However, no other study was excluded from this review based on its design alone.

Oral hygiene care using CHX is currently not actively recommended. Some studies have reported that use of CHX for oral hygiene care is associated with adverse outcomes, such as mortality. For example, a meta-analysis on the efficacy of oral hygiene care without CHX in non-cardiac surgery patients showed that the use of CHX increased mortality; however, the result was not significant [46]. Price et al. analyzed the safety of CHX and compared the following four groups in their NMA: selective digestive decontamination group, selective oropharyngeal decontamination group, oropharyngeal chlorhexidine (CHX group), and standard care or placebo (control group) [47]. Pairwise meta-analysis showed that the CHX group had a significantly increased risk of mortality compared with the control group. In addition, NMA of the risk of mortality showed that the CHX group ranked lower than the control group [47]. In light of these findings, use of CHX for oral hygiene care is not actively recommended in recent guidelines and consensus papers [10,48]. Therefore, use of CHX for oral hygiene care may be essentially ranked lower than that in our results and may not be recommended in clinical practice.

This study has some limitations. First, despite the evaluation of heterogeneity and consistency in this study, some factors may have affected the results of the NMA. For example, the frequency of oral hygiene care per day varied between studies; however, we did not categorize the differences in frequency. Nevertheless, we believe that our inclusion criteria, such as the definition of intervention duration and the exclusion of complex interventions, could minimize potential heterogeneity. Moreover, the frequencies of the interventions in the included studies were similar (approximately 2–3 times per day). Second, the number of studies that were directly compared may be insufficient. Although sensitivity analysis should be conducted if the VAP assessor in a study was not blinded, we could not conduct sensitivity analyses because the network plot was not connected when the applicable study was excluded. In addition, of the 12 direct comparisons conducted, the most common comparison was three cases. Third, most of the included studies had 'some concerns' regarding the risk of bias, mainly because they did not have a study protocol. Moreover, of the 12 studies, one had a high overall risk of bias because the VAP assessor was not blinded. Finally, the confidence ratings using CINeMA were categorized as low or very low for each comparison. While these ratings were primarily due to issues with protocol reporting and the limited number of studies, we consider these issues could be addressed with future research. However, the remaining concern about imprecision is a limitation of the present study.

## Clinical implications

We suggest that the use of a toothbrush for oral hygiene care be included in the recommended clinical practice for preventing VAP. Oral hygiene care using a toothbrush consistently ranked highly for effectiveness in our analyses. Moreover, while a toothbrush is a low-cost option with high feasibility for implementation, concerns have been raised regarding potential adverse outcomes associated with the use of CHX. Overall, these data indicate that toothbrush-based oral hygiene care is a promising approach to preventing VAP in ICU patients.

## Research implications

Further research on oral hygiene care methods involving brushing is needed. For example, exploring the optimal brushing frequency and timing to prevent VAP is a critical area for future investigation. Future research should also determine the appropriate softness of tooth-brush bristles and the ideal toothbrush size for the effective and safe removal of dental plaque. Identifying the optimal techniques to eliminate contaminants after brushing, such as wiping and rinsing, is also a priority. Moreover, several studies were excluded from this analysis due to the intervention duration being shorter than the period of mechanical ventilation. Future studies should consider extending the focus beyond early-onset VAP to cases occurring throughout the entire duration of mechanical ventilation, which may yield insights into favorable patient outcomes over the long term. Finally, there is limited research on oral cavity conditions, including dental caries, tooth loss, and dry mouth. These factors may be relevant to the interpretation of the data on VAP and should be incorporated into future research designs.

## Conclusions

The present NMA indicated that brushing combined with chlorhexidine 0.12% may be an effective intervention for the prevention of VAP in ICU patients. In addition, the results of this study show that oral hygiene care that includes brushing may be recommended as the optimal method for preventing VAP in clinical settings. However, as the number of studies included in this NMA is small, further research is needed to obtain more accurate results regarding the optimal oral hygiene care methods for preventing VAP in ICU patients.

## Supporting information

**S1 Checklist. PRISMA 2020 checklist.**
(DOCX)

**S1 Fig. Forest plot for direct and indirect comparisons of treatment effects using the node-splitting method.**
(DOCX)

**S2 Fig. Network Plot with CINeMA.**
(DOCX)

**S3 Fig. Funnel plot.**
(DOCX)

**S1 Text. PubMed search strategy.**
(DOCX)

**S1 Table. All studies identified in the literature search.**
(XLSX)

**S2 Table. Evaluation of each domain and parameter in the Risk of Bias (RoB) assessment.**
(XLSX)

**S3 Table. Assessment of quality of evidence using CINeMA.**
(DOCX)

**S4 Table. Extracted data.**
(XLSX)

## Author Contributions

**Conceptualization:** Sachika Yamakita, Takeshi Unoki, Sachi Niiyama, Eri Natsuhori, Junpei Haruna, Tomoki Kuribara.

**Data curation:** Sachika Yamakita, Takeshi Unoki, Sachi Niiyama, Eri Natsuhori, Junpei Haruna, Tomoki Kuribara.

**Formal analysis:** Sachika Yamakita, Takeshi Unoki.

**Funding acquisition:** Takeshi Unoki.

**Investigation:** Sachika Yamakita, Takeshi Unoki, Sachi Niiyama, Eri Natsuhori, Junpei Haruna, Tomoki Kuribara.

**Methodology:** Sachika Yamakita, Takeshi Unoki.

**Project administration:** Takeshi Unoki.

**Supervision:** Takeshi Unoki.

**Validation:** Sachika Yamakita, Takeshi Unoki.

**Visualization:** Sachika Yamakita, Takeshi Unoki.

**Writing – original draft:** Sachika Yamakita, Takeshi Unoki.

**Writing – review & editing:** Sachika Yamakita, Takeshi Unoki, Sachi Niiyama, Eri Natsuhori, Junpei Haruna, Tomoki Kuribara.

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
