## [Decision Letter · Decision Letter 0]

22 Jul 2024

PONE-D-24-19045Comparative efficacy of various oral hygiene care methods in preventing ventilator-associated pneumonia in critically ill patients: A systematic review and network meta-analysisPLOS ONE

Dear Dr. Unoki,

Thank you for submitting your manuscript to PLOS ONE. After careful consideration, we feel that it has merit but does not fully meet PLOS ONE’s publication criteria as it currently stands. Therefore, we invite you to submit a revised version of the manuscript that addresses the points raised during the review process. Include systematic review registration number on PROSPERO or another base;Indicate results with quantitative data;Indicate the p-value in the results;Explain the methodology in more depth.

We look forward to receiving your revised manuscript.

Kind regards,

Maria Giulia Nosotti, Master's Degree

Academic Editor

PLOS ONE

Additional Editor Comments (if provided):

Reviewers' comments:

Reviewer's Responses to Questions

**Comments to the Author**

1. Is the manuscript technically sound, and do the data support the conclusions?

Reviewer #1: Yes

Reviewer #2: Yes

2. Has the statistical analysis been performed appropriately and rigorously? 

Reviewer #1: Yes

Reviewer #2: No

3. Have the authors made all data underlying the findings in their manuscript fully available?

Reviewer #1: Yes

Reviewer #2: No

4. Is the manuscript presented in an intelligible fashion and written in standard English?

Reviewer #1: No

Reviewer #2: Yes

5. Review Comments to the Author

Reviewer #1: Abstract

Include systematic review registration number on PROSPERO or another base.Indicate results with quantitative data;explain in detail the methodology used;

Methods

- Specify the adopted PICO criterion or similar;

- Explain how the manual search was performed;

The search was carried out until July 2022. Update the search period; this part is confusing. Keep only the final search period.

- Report on using the ROB scale in the summary as well.

- Wouldn't it be essential to consider using GRADE?

Synthesis of results

- When indicating that there was a significant difference, enter the p-value.

Discussion

This section needs to be implemented;

Include hypotheses in the study in the methodology and respond in the discussion;

indicate trends for future studies;

Address suggestions for clinical practice;

Figures

the meta-analysis chart indicates the studies involved

or include the origin of the data in the legend;

In each result figure, include the studies

engaged in the analysis in the figure caption, for example.

Reviewer #2: The correlation between arms within the same study needs to be adjusted to avoid bias and incorrect estimates.

Is there a burn-in period used? How many iterations were discarded?

Specify the number of chains run in parallel, as convergence across multiple chains is more reliable.

Perhaps it is good to use multiple diagnostics to confirm convergence.

6. PLOS authors have the option to publish the peer review history of their article (what does this mean?). If published, this will include your full peer review and any attached files.

Reviewer #1: No

Reviewer #2: No

---

## [Author Response · Author response to Decision Letter 0]

27 Sep 2024

Responses to the Reviewers

Thank you for reviewing our manuscript and providing valuable suggestions. We have addressed all the reviewers’ comments and provided our point-by-point responses below.

Comments from Academic Editor

We invite you to submit a revised version of the manuscript that addresses the points raised during the review process.

Include systematic review registration number on PROSPERO or another base;

Indicate results with quantitative data;

Indicate the p-value in the results;

Explain the methodology in more depth.

Thank you for your consideration. We have revised our manuscript and incorporated the requested information, to the best of our ability.

Comments from Reviewer #1

Abstract

Include systematic review registration number on PROSPERO or another base.

We apologize for not including the protocol registration number on PROSPERO in this section. We have added the information to the Abstract accordingly. 

Page 2, Line 32 - L 33

The review protocol was registered in PROSPERO (registration number: CRD42022333270). 

Indicate results with quantitative data;

We added the description below.

Page 2, Line 33 - L 40

Thirteen randomized controlled trials were included in the qualitative synthesis and twelve randomized controlled trials (2395 participants) were included in the network meta-analysis. Over 50% of the included studies were conducted in medical-surgical intensive care units. Ten treatments were analyzed and 12 pairwise comparisons were conducted in the 12 included studies. Analysis using surface under the cumulative ranking curves revealed that brushing combined with chlorhexidine 0.12% was most likely the optimal intervention for preventing ventilator-associated pneumonia (88.4%), followed by the use of chlorhexidine 0.12% alone (76.1%), and brushing alone (73.2%).

explain in detail the methodology used;

We have provided a more detailed explanation of the methodology, as follows: 

Page 2, Line 25 - L 33

A literature search of three representative databases was conducted. We only analyzed parallel randomized controlled trials conducted to analyze the use antiseptics or toothbrushes in oral hygiene care for adult patients undergoing invasive mechanical ventilation in the intensive care unit. The outcome measure was the incidence of ventilator-associated pneumonia. Bias risk was assessed using the Cochrane Risk of Bias 2 tool, and the confidence in the evidence was evaluated using the CINeMA approach. Statistical analyses were performed using R 4.2.0., GeMTC package, and JAGS 4.3.1. The review protocol was registered in PROSPERO (registration number: CRD42022333270). 

Methods

Specify the adopted PICO criterion or similar;

We apologize for the unclear description. The study eligibility criteria were described on Page 4, Line 91 - P5 L 105 in the original version of the manuscript. However, we have revised these criteria based on PICO, as follows:

Page 4, Line 79 - L 107

Eligibility criteria

Population

Participants were adult patients who were admitted to the ICU and underwent invasive mechanical ventilation.

Intervention

Oral hygiene care using antiseptics and/or tooth brushing was provided continuously throughout the duration of mechanical ventilation. Studies were excluded if any of the following criteria applied: (1) oral hygiene care included the use of antibiotics; (2) patients underwent an intervention that directly involved the larynx and/or pharynx; (3) oral hygiene care was provided temporarily and/or not continuously (e.g., only once after tracheal intubation, for three days only regardless of whether ventilation was required for longer); (4) the intervention involved the use of complex care bundles in which the differences between the intervention and control groups included factors other than tooth brushing, rinsing the mouth, or applying gel (e.g., differences in body position, modification of tube cuff pressure). 

Regarding eligible treatments included in the treatment network, CHX was stratified according to its concentration. Manual and electronic brushing were considered the same treatment. Different frequencies of the same intervention were considered the same node. Different types of antiseptics were considered the same treatment because stratification of antiseptics according to type led to several categories of treatments, which may affect the network geometry.

Comparison and Control

Comparisons were usual care (e.g., using only normal saline), placebo, and oral hygiene intervention care if the studies met the inclusion criteria. In this NMA, the control was described as placebo or usual care, with no tooth brushing or use of antiseptics (e.g., using only normal saline),

Outcome and Study Design

The outcome was the incidence of VAP. The definition of VAP was based on the definition used in each study. Only parallel randomized controlled trials were included. Articles without full text, letters without reported data, and reviews were excluded.

Explain how the manual search was performed;

We have added the following description:

Page 5, Line 115 - L 117

Manual searches were also conducted to identify additional relevant studies. We manually checked all the references lists of the relevant systematic reviews to identify any additional studies.

The search was carried out until July 2022. Update the search period; this part is confusing. Keep only the final search period.

We have revised the description as follows:

Page 5, Line 114-115

The databases were searched from inception to March 24, 2024

Report on using the ROB scale in the summary as well.

We have added the sentence below in the Abstract.

.

Page 2, Line 29 - L 31

Bias risk was assessed using the Cochrane Risk of Bias 2 tool, and the confidence in the evidence was evaluated using the CINeMA approach.

Wouldn't it be essential to consider using GRADE?

We have conducted an assessment of the confidence in the results using the CINeMA approach. We have added the following description in the Methods and Results sections. The results of the assessment with CINeMA are shown in S3 Fig and S3 Text.

Page 8, Line 182 - L 195

Methods

Assessment of the quality of the evidence using Confidence in Network meta-analysis (CINeMA)

Confidence in the evidence was evaluated using the CINeMA approach [27,28]. CINeMA is broadly based on the GRADE framework and covers 6 domains: (i) within-study bias (referring to the impact of risk of bias in the included studies), (ii) reporting bias (referring to publication and other reporting biases), (iii) indirectness, (iv) imprecision, (v) heterogeneity, and (vi) incoherence. Each of these domains can assessed according to three levels of concern (no concerns, some concerns, or major concerns) and the reviewer’s input is required at the study level for within-study bias and indirectness. Judgments across domains are summarized to obtain 4 levels of confidence for each relative treatment effect, corresponding to the usual GRADE assessments of very low, low, moderate, or high. Two authors (SY and TU) independently made judgments about the quality of the evidence. Conflicts were resolved through discussions and consensus. These assessments were conducted using the CINeMA web application [29].

Results

Page 16, Line 61 - L 65

However, the confidence rating with CINeMA was low or very low in each comparison. The main reasons for downgrading were related to within-study bias, with many studies rated as having 'Some concerns' due to the absence of protocols. Additionally, concerns regarding imprecision were noted due to the small number of included studies. The results of the CINeMA assessment are provided in the Supporting Information (S3 Fig and S3 Text).

Synthesis of results

When indicating that there was a significant difference, enter the p-value.

We have stated the p-values where available. However, for some analyses, p-values were not generated due to the nature of the analytical methods used.

Discussion

This section needs to be implemented;

We apologize for this oversight. We have added the required sections.

Include hypotheses in the study in the methodology and respond in the discussion;

We hypothesized that oral hygiene care using toothbrushing is the most effective method for preventing VAP in patients on ICU. This is based on the characteristics of the toothbrush, which physically removes dental plaque, and on previous studies demonstrating that brushing alone is the best intervention for preventing VAP. Notably, the key advantages of toothbrushes are that they are low-cost and a feasible intervention. We added these points to the Introduction on Page 3, Line 56 - L 67.

Moreover, in the Discussion section, we have identified reasons for differences between our findings and those of previous studies and discussed the optimal oral hygiene care in clinical settings, while ensuring that the connection between the hypothesis and discussion was carefully maintained.

Page 3, Line 56 - L 70

Toothbrushing is recognized as an important component of oral hygiene care in critically ill patients allowing physical removal of dental plaque that is a potential nidus of infection [10]. A network meta-analysis (NMA) comparing 16 oral hygiene interventions has demonstrated that toothbrushing alone is the most effective approach [11]. However, important limitations of the NMA are the inclusion of studies with a duration of intervention that was shorter than the use of mechanical ventilation and of studies including interventions directed to the pharynx rather than the oral cavity [8,12–14]. Furthermore, toothbrushes are a low-cost intervention with high feasibility for implementation. In surveys investigating oral care methods used in ICUs, toothbrushes are frequently utilized [15,16]. Thus, toothbrushes are considered a relatively easy-to-apply and effective method for preventing VAP in clinical settings.

Therefore, we hypothesized that oral hygiene care involving toothbrushing would be the most effective method for preventing VAP and conducted an NMA to compare the efficacy of various oral hygiene care methods, including the use of antiseptics and toothbrushing, on the incidence of VAP in adult patients undergoing invasive mechanical ventilation. In addition, we ranked these methods by effectiveness for practical consideration.

indicate trends for future studies;

We revised the text to identify trends for future research, as follows:

Page 24, Line 197 - L 209

Further research on oral hygiene care methods involving brushing is needed. For example, exploring the optimal brushing frequency and timing to prevent VAP is a critical area for future investigation. Future research should also determine the appropriate softness of toothbrush bristles and the ideal toothbrush size for the effective and safe removal of dental plaque. Identifying the optimal techniques to eliminate contaminants after brushing, such as wiping and rinsing, is also a priority. Moreover, several studies were excluded from this analysis due to the intervention duration being shorter than the period of mechanical ventilation. Future studies should consider extending the focus beyond early-onset VAP to cases occurring throughout the entire duration of mechanical ventilation, which may yield insights into favorable patient outcomes over the long term. Finally, there is limited research on oral cavity conditions, including dental caries, tooth loss, and dry mouth. These factors may be relevant to the interpretation of the data on VAP and should be incorporated into future research designs.

Address suggestions for clinical practice;

We have added the following section on clinical implications to the Discussion: 

Page 23, Line 189 - L 195

Clinical implications

We suggest that the use of a toothbrush for oral hygiene care be included in the recommended clinical practice for preventing VAP. Oral hygiene care using a toothbrush consistently ranked highly for effectiveness in our analyses. Moreover, while a toothbrush is a low-cost option with high feasibility for implementation, concerns have been raised regarding potential adverse outcomes associated with the use of CHX. Overall, these data indicate that toothbrush-based oral hygiene care is a promising approach to preventing VAP in ICU patients.

Figures

the meta-analysis chart indicates the studies involved or include the origin of the data in the legend; In each result figure, include the studies engaged in the analysis in the figure caption, for example.

When the caption can be incorporated within the figure, we have added the reference numbers. In more complex figures, the reference numbers for the studies providing data for each intervention are shown in the corresponding legend.

Fig 2 Legend (added)

The sources of data for each intervention: Br [32,33,35,42], BrBicarbonate [32,35], BrCHX012 [31,34,40,42], BrCHX02 [32,38,39], BrCHX2 [33,39], BrListerine [35], CHX012 [31,34,40], CHX02 [30,36,38], CHX2 [37], Ctr [30,36,37].

 Fig 4

Comments from Reviewer #2

The correlation between arms within the same study needs to be adjusted to avoid bias and incorrect estimates. 

As you have noted, our included studies comprise two multi-arm trials. We have reanalyzed the data using powerAdjust to adjust for correlations between arms within the same study. In addition, we have updated the Methods section and revised the numerical results accordingly. There were no changes in the SUCRA rankings as a result of the adjustment for multi-arm trials. Moreover, we added the description below in the method section.

Page 6, Line 149 - L 154

For the analysis of multi-arm trials with more than two intervention arms, each study was included as a two-arm comparison series in the dataset. Thus, we obtained the treatment effects for each intervention in relation to every comparator in the multi-arm. We adjusted the data using the "powerAdjust" argument, an option available in the GeMTC package, setting the weight for multi-arm trials at 0.7 for correlations between arms within the same study [21-23].

Is there a burn-in period used? How many iterations were discarded?

Specify the number of chains run in parallel, as convergence across multiple chains is more reliable.

We discarded the first 5,000 iterations as a burn-in period. The number of chains run in parallel is four. We have added these details in the revised manuscript.

Page 7, Line 155 - L 157

To ensure convergence of the Markov chains, we discarded the first 5,000 iterations as the burn-in period and ran four chains in parallel.

Perhaps it is good to use multiple diagnostics to confirm convergence.

We have used two diagnostics to confirm convergence. We have revised the description as follows:

Page 7, Line 154 - L 159

The analysis was performed using the Markov-Chain Monte-Carlo (MCMC) method. The NMA was performed using a random effects model. To ensure convergence of the Markov chains, we discarded the first 5,000 iterations as the burn-in period and ran four chains in parallel. The convergence was assessed using two diagnostics, the visual checking of trace plots and the potential scale reduction factor (PSRF) of Gelman-Rubin.

---

## [Editor Report · Decision Letter 1]

1 Oct 2024

PONE-D-24-19045R1Comparative efficacy of various oral hygiene care methods in preventing ventilator-associated pneumonia in critically ill patients: A systematic review and network meta-analysisPLOS ONE

Dear Dr. Unoki,

Thank you for submitting your manuscript to PLOS ONE. After careful consideration, we feel that it has merit but does not fully meet PLOS ONE’s publication criteria as it currently stands. Therefore, we invite you to submit a revised version of the manuscript that addresses the points raised during the review process. Abstract

- Include systematic review registration number on PROSPERO or another base.

- Indicate results with quantitative data;

Methods

- Specify the adopted PICO criterion or similar;

- Explain how the manual search was performed;

The search was carried out until July 2022. Update the search period; this part is confusing. Keep only the final search period.

- Report on using the ROB scale in the summary as well.

- Wouldn't it be essential to consider using GRADE?

Synthesis of results

- When indicating that there was a significant difference, enter the p-value.

Discussion

This section needs to be implemented;

Include hypotheses in the study in the methodology and respond in the discussion;

indicate trends for future studies;

Address suggestions for clinical practice;

Figures

the meta-analysis chart indicates the studies involved

or include the origin of the data in the legend;

In each result figure, include the studies

engaged in the analysis in the figure caption, for example.

We look forward to receiving your revised manuscript.

Kind regards,

Maria Giulia Nosotti, Master's Degree

Academic Editor

PLOS ONE

---

## [Author Response · Author response to Decision Letter 1]

15 Oct 2024

Responses to the Reviewers

Thank you for reviewing our manuscript and providing valuable suggestions. We have addressed all the reviewers’ comments and provided our point-by-point responses below.

Comments from Academic Editor

We invite you to submit a revised version of the manuscript that addresses the points raised during the review process.

Include systematic review registration number on PROSPERO or another base;

Indicate results with quantitative data;

Indicate the p-value in the results;

Explain the methodology in more depth.

Thank you for your consideration. We have revised our manuscript and incorporated the requested information, to the best of our ability.

Comments from Reviewer #1

Abstract

Include systematic review registration number on PROSPERO or another base.

We apologize for not including the protocol registration number on PROSPERO in this section. We have added the information to the Abstract accordingly. 

Page 2, Line 32 - L 33

The review protocol was registered in PROSPERO (registration number: CRD42022333270). 

Indicate results with quantitative data;

We added the description below.

Page 2, Line 33 - L 40

Thirteen randomized controlled trials were included in the qualitative synthesis and twelve randomized controlled trials (2395 participants) were included in the network meta-analysis. Over 50% of the included studies were conducted in medical-surgical intensive care units. Ten treatments were analyzed and 12 pairwise comparisons were conducted in the 12 included studies. Analysis using surface under the cumulative ranking curves revealed that brushing combined with chlorhexidine 0.12% was most likely the optimal intervention for preventing ventilator-associated pneumonia (88.4%), followed by the use of chlorhexidine 0.12% alone (76.1%), and brushing alone (73.2%).

explain in detail the methodology used;

We have provided a more detailed explanation of the methodology, as follows: 

Page 2, Line 25 - L 33

A literature search of three representative databases was conducted. We only analyzed parallel randomized controlled trials conducted to analyze the use antiseptics or toothbrushes in oral hygiene care for adult patients undergoing invasive mechanical ventilation in the intensive care unit. The outcome measure was the incidence of ventilator-associated pneumonia. Bias risk was assessed using the Cochrane Risk of Bias 2 tool, and the confidence in the evidence was evaluated using the CINeMA approach. Statistical analyses were performed using R 4.2.0., GeMTC package, and JAGS 4.3.1. The review protocol was registered in PROSPERO (registration number: CRD42022333270). 

Methods

Specify the adopted PICO criterion or similar;

We apologize for the unclear description. The study eligibility criteria were described on Page 4, Line 91 - P5 L 105 in the original version of the manuscript. However, we have revised these criteria based on PICO, as follows:

Page 4, Line 79 - L 107

Eligibility criteria

Population

Participants were adult patients who were admitted to the ICU and underwent invasive mechanical ventilation.

Intervention

Oral hygiene care using antiseptics and/or tooth brushing was provided continuously throughout the duration of mechanical ventilation. Studies were excluded if any of the following criteria applied: (1) oral hygiene care included the use of antibiotics; (2) patients underwent an intervention that directly involved the larynx and/or pharynx; (3) oral hygiene care was provided temporarily and/or not continuously (e.g., only once after tracheal intubation, for three days only regardless of whether ventilation was required for longer); (4) the intervention involved the use of complex care bundles in which the differences between the intervention and control groups included factors other than tooth brushing, rinsing the mouth, or applying gel (e.g., differences in body position, modification of tube cuff pressure). 

Regarding eligible treatments included in the treatment network, CHX was stratified according to its concentration. Manual and electronic brushing were considered the same treatment. Different frequencies of the same intervention were considered the same node. Different types of antiseptics were considered the same treatment because stratification of antiseptics according to type led to several categories of treatments, which may affect the network geometry.

Comparison and Control

Comparisons were usual care (e.g., using only normal saline), placebo, and oral hygiene intervention care if the studies met the inclusion criteria. In this NMA, the control was described as placebo or usual care, with no tooth brushing or use of antiseptics (e.g., using only normal saline),

Outcome and Study Design

The outcome was the incidence of VAP. The definition of VAP was based on the definition used in each study. Only parallel randomized controlled trials were included. Articles without full text, letters without reported data, and reviews were excluded.

Explain how the manual search was performed;

We have added the following description:

Page 5, Line 115 - L 117

Manual searches were also conducted to identify additional relevant studies. We manually checked all the references lists of the relevant systematic reviews to identify any additional studies.

The search was carried out until July 2022. Update the search period; this part is confusing. Keep only the final search period.

We have revised the description as follows:

Page 5, Line 114-115

The databases were searched from inception to March 24, 2024

Report on using the ROB scale in the summary as well.

We have added the sentence below in the Abstract.

.

Page 2, Line 29 - L 31

Bias risk was assessed using the Cochrane Risk of Bias 2 tool, and the confidence in the evidence was evaluated using the CINeMA approach.

Wouldn't it be essential to consider using GRADE?

We have conducted an assessment of the confidence in the results using the CINeMA approach. We have added the following description in the Methods and Results sections. The results of the assessment with CINeMA are shown in S3 Fig and S3 Text.

Page 8, Line 182 - L 195

Methods

Assessment of the quality of the evidence using Confidence in Network meta-analysis (CINeMA)

Confidence in the evidence was evaluated using the CINeMA approach [27,28]. CINeMA is broadly based on the GRADE framework and covers 6 domains: (i) within-study bias (referring to the impact of risk of bias in the included studies), (ii) reporting bias (referring to publication and other reporting biases), (iii) indirectness, (iv) imprecision, (v) heterogeneity, and (vi) incoherence. Each of these domains can assessed according to three levels of concern (no concerns, some concerns, or major concerns) and the reviewer’s input is required at the study level for within-study bias and indirectness. Judgments across domains are summarized to obtain 4 levels of confidence for each relative treatment effect, corresponding to the usual GRADE assessments of very low, low, moderate, or high. Two authors (SY and TU) independently made judgments about the quality of the evidence. Conflicts were resolved through discussions and consensus. These assessments were conducted using the CINeMA web application [29].

Results

Page 16, Line 61 - L 65

However, the confidence rating with CINeMA was low or very low in each comparison. The main reasons for downgrading were related to within-study bias, with many studies rated as having 'Some concerns' due to the absence of protocols. Additionally, concerns regarding imprecision were noted due to the small number of included studies. The results of the CINeMA assessment are provided in the Supporting Information (S3 Fig and S3 Text).

Synthesis of results

When indicating that there was a significant difference, enter the p-value.

We have stated the p-values where available. However, for some analyses, p-values were not generated due to the nature of the analytical methods used.

Discussion

This section needs to be implemented;

We apologize for this oversight. We have added the required sections.

Include hypotheses in the study in the methodology and respond in the discussion;

We hypothesized that oral hygiene care using toothbrushing is the most effective method for preventing VAP in patients on ICU. This is based on the characteristics of the toothbrush, which physically removes dental plaque, and on previous studies demonstrating that brushing alone is the best intervention for preventing VAP. Notably, the key advantages of toothbrushes are that they are low-cost and a feasible intervention. We added these points to the Introduction on Page 3, Line 56 - L 67.

Moreover, in the Discussion section, we have identified reasons for differences between our findings and those of previous studies and discussed the optimal oral hygiene care in clinical settings, while ensuring that the connection between the hypothesis and discussion was carefully maintained.

Page 3, Line 56 - L 70

Toothbrushing is recognized as an important component of oral hygiene care in critically ill patients allowing physical removal of dental plaque that is a potential nidus of infection [10]. A network meta-analysis (NMA) comparing 16 oral hygiene interventions has demonstrated that toothbrushing alone is the most effective approach [11]. However, important limitations of the NMA are the inclusion of studies with a duration of intervention that was shorter than the use of mechanical ventilation and of studies including interventions directed to the pharynx rather than the oral cavity [8,12–14]. Furthermore, toothbrushes are a low-cost intervention with high feasibility for implementation. In surveys investigating oral care methods used in ICUs, toothbrushes are frequently utilized [15,16]. Thus, toothbrushes are considered a relatively easy-to-apply and effective method for preventing VAP in clinical settings.

Therefore, we hypothesized that oral hygiene care involving toothbrushing would be the most effective method for preventing VAP and conducted an NMA to compare the efficacy of various oral hygiene care methods, including the use of antiseptics and toothbrushing, on the incidence of VAP in adult patients undergoing invasive mechanical ventilation. In addition, we ranked these methods by effectiveness for practical consideration.

indicate trends for future studies;

We revised the text to identify trends for future research, as follows:

Page 24, Line 197 - L 209

Further research on oral hygiene care methods involving brushing is needed. For example, exploring the optimal brushing frequency and timing to prevent VAP is a critical area for future investigation. Future research should also determine the appropriate softness of toothbrush bristles and the ideal toothbrush size for the effective and safe removal of dental plaque. Identifying the optimal techniques to eliminate contaminants after brushing, such as wiping and rinsing, is also a priority. Moreover, several studies were excluded from this analysis due to the intervention duration being shorter than the period of mechanical ventilation. Future studies should consider extending the focus beyond early-onset VAP to cases occurring throughout the entire duration of mechanical ventilation, which may yield insights into favorable patient outcomes over the long term. Finally, there is limited research on oral cavity conditions, including dental caries, tooth loss, and dry mouth. These factors may be relevant to the interpretation of the data on VAP and should be incorporated into future research designs.

Address suggestions for clinical practice;

We have added the following section on clinical implications to the Discussion: 

Page 23, Line 189 - L 195

Clinical implications

We suggest that the use of a toothbrush for oral hygiene care be included in the recommended clinical practice for preventing VAP. Oral hygiene care using a toothbrush consistently ranked highly for effectiveness in our analyses. Moreover, while a toothbrush is a low-cost option with high feasibility for implementation, concerns have been raised regarding potential adverse outcomes associated with the use of CHX. Overall, these data indicate that toothbrush-based oral hygiene care is a promising approach to preventing VAP in ICU patients.

Figures

the meta-analysis chart indicates the studies involved or include the origin of the data in the legend; In each result figure, include the studies engaged in the analysis in the figure caption, for example.

When the caption can be incorporated within the figure, we have added the reference numbers. In more complex figures, the reference numbers for the studies providing data for each intervention are shown in the corresponding legend.

Fig 2 Legend (added)

The sources of data for each intervention: Br [32,33,35,42], BrBicarbonate [32,35], BrCHX012 [31,34,40,42], BrCHX02 [32,38,39], BrCHX2 [33,39], BrListerine [35], CHX012 [31,34,40], CHX02 [30,36,38], CHX2 [37], Ctr [30,36,37].

 Fig 4

Comments from Reviewer #2

The correlation between arms within the same study needs to be adjusted to avoid bias and incorrect estimates. 

As you have noted, our included studies comprise two multi-arm trials. We have reanalyzed the data using powerAdjust to adjust for correlations between arms within the same study. In addition, we have updated the Methods section and revised the numerical results accordingly. There were no changes in the SUCRA rankings as a result of the adjustment for multi-arm trials. Moreover, we added the description below in the method section.

Page 6, Line 149 - L 154

For the analysis of multi-arm trials with more than two intervention arms, each study was included as a two-arm comparison series in the dataset. Thus, we obtained the treatment effects for each intervention in relation to every comparator in the multi-arm. We adjusted the data using the "powerAdjust" argument, an option available in the GeMTC package, setting the weight for multi-arm trials at 0.7 for correlations between arms within the same study [21-23].

Is there a burn-in period used? How many iterations were discarded?

Specify the number of chains run in parallel, as convergence across multiple chains is more reliable.

We discarded the first 5,000 iterations as a burn-in period. The number of chains run in parallel is four. We have added these details in the revised manuscript.

Page 7, Line 155 - L 157

To ensure convergence of the Markov chains, we discarded the first 5,000 iterations as the burn-in period and ran four chains in parallel.

Perhaps it is good to use multiple diagnostics to confirm convergence.

We have used two diagnostics to confirm convergence. We have revised the description as follows:

Page 7, Line 154 - L 159

The analysis was performed using the Markov-Chain Monte-Carlo (MCMC) method. The NMA was performed using a random effects model. To ensure convergence of the Markov chains, we discarded the first 5,000 iterations as the burn-in period and ran four chains in parallel. The convergence was assessed using two diagnostics, the visual checking of trace plots and the potential scale reduction factor (PSRF) of Gelman-Rubin.

---

## [Editor Report · Decision Letter 2]

18 Oct 2024

Comparative efficacy of various oral hygiene care methods in preventing ventilator-associated pneumonia in critically ill patients: A systematic review and network meta-analysis

PONE-D-24-19045R2

Dear Dr. Unoki,

We’re pleased to inform you that your manuscript has been judged scientifically suitable for publication and will be formally accepted for publication once it meets all outstanding technical requirements.

Kind regards,

Maria Giulia Nosotti, Master's Degree

Academic Editor

PLOS ONE
---

## [Editor Report · Acceptance letter]

14 Nov 2024

PONE-D-24-19045R2 

PLOS ONE

Dear Dr. Unoki, 

I'm pleased to inform you that your manuscript has been deemed suitable for publication in PLOS ONE. Congratulations! Your manuscript is now being handed over to our production team.

Kind regards, 

on behalf of

Dr. Maria Giulia Nosotti 

Academic Editor

PLOS ONE